# Reaching Nirvana: Maximizing the Margin in Both Euclidean and Angular Spaces for Deep Neural Network Classification

## Abstract

The classification loss functions used in deep neural network classifiers can be grouped into two categories based on maximizing the margin in either Euclidean or angular spaces. Euclidean distances between sample vectors are used during classification for the methods maximizing the margin in Euclidean spaces whereas the Cosine similarity distance is used during the testing stage for the methods maximizing margin in the angular spaces. This paper introduces a novel classification loss that maximizes the margin in both the Euclidean and angular spaces at the same time. This way, the Euclidean and Cosine distances will produce similar and consistent results and complement each other, which will in turn improve the accuracies. The proposed loss function enforces the samples of classes to cluster around the centers that represent them. The centers approximating classes are chosen from the boundary of a hypersphere, and the pairwise distances between class centers are always equivalent. This restriction corresponds to choosing centers from the vertices of a regular simplex. There is not any hyperparameter that must be set by the user in the proposed loss function, therefore the use of the proposed method is extremely easy for classical classification problems. Moreover, since the class samples are compactly clustered around their corresponding means, the proposed classifier is also very suitable for open set recognition problems where test samples can come from the unknown classes that are not seen in the training phase. Experimental studies show that the proposed method achieves the state-of-the-art accuracies on open set recognition despite its simplicity.

## 1 Introduction

Deep neural network classifiers have been dominating many fields including computer vision by achieving state-of-the-art accuracies in many tasks such as visual object, activity, face and scene classification. Therefore, new deep neural network architectures and different classification losses have been constantly developing. The softmax loss function is the most common function used for classification in deep neural network classifiers. Although the softmax loss yields satisfactory accuracies for general object classification problems, its performance for discrimination of the instances coming from the same class categories (e.g., face recognition) or open set recognition (a classification scenario that allows the test samples to come from the unknown classes) is not satisfactory. The performance decrease is typically attributed to two factors: there is no mechanism for enforcing large-margin between classes and the softmax does not attempt to minimize the within-class scatter which is crucial for the success in open set recognition problems.

To improve the classification accuracies of the deep neural network classifiers, many researchers focused on maximizing the margin between classes. The recent methods can be roughly divided into

Submitted to 36th Conference on Neural Information Processing Systems (NeurIPS 2022). Do not distribute.

two categories based on maximizing the margin in either Euclidean or angular spaces. The methods targeting margin maximization in the Euclidean spaces attempt to minimize the Euclidean distances among the samples coming from the same classes and maximize the distances among the samples coming from different classes. Euclidean distances are used during testing stage after the network is trained. In contrast, the methods that maximize the margin in the angular spaces use the cosine distances for classification.

To maximize the margin in Euclidean space, Wen et al. [1, 2] combined the softmax loss function with the center loss for face recognition. Center loss reduces the within-class variations by minimizing the distances between the individual face class samples and their corresponding class centers. The resulting method significantly improves the accuracies over the method using softmax alone in the context of face recognition. A variant of the center loss called the contrastive center loss [3] minimizes the Euclidean distances between the samples and their corresponding class centers and maximizes the distances between samples and the centers of the rival (non-corresponding) classes. Zhang et al. [4] combined the range loss with the softmax loss to maximize the margin in the Euclidean spaces. Wei et al. [5] combined softmax loss and center loss functions with the minimum margin loss where the minimum margin loss enforces all class center pairs to have a distance larger than a specified threshold. Deng et al. [6] introduced a method using softmax loss function with the marginal loss to create compact and well separated classes in Euclidean space. Cevikalp et al. [7] proposed a deep neural network based open set recognition method that returns compact class acceptance regions for each known class. In this framework, hinge loss and polyhedral conic functions are used for the between-class separation. The methods using Contrastive loss minimize the Euclidean distance of the positive sample pairs and penalize the negative pairs that have a distance smaller than a given margin threshold. In a similar manner, [8, 9, 10, 11] employ triplet loss function that used a positive sample, a negative sample and an anchor. An anchor is also a positive sample, thus the within-class compactness is achieved by minimizing the Euclidean distances between the anchor and positive samples whereas the distances between anchor and negative samples are maximized for between-class separation. Although methods using both contrastive and triplet loss functions return compact decision boundaries, they have limitations in the sense that the number of sample pairs or triplets grows quadratically (cubicly) compared to the total number of samples, which results in slow convergence and instability. A careful sampling/mining of data is required to avoid this problem. Overall, the majority of the methods maximizing margin in the Euclidean spaces have shortcomings in a way that they are too complex since the user has to set many weighting and margin parameters. This is due to the fact that the main classification loss functions include many terms that needs to be properly weighted. Furthermore, many of these methods are not suitable for open set recognition problems since they do not return compact acceptance regions for classes.

The methods that enlarge the margin in the angular spaces typically revise the classical softmax loss functions to maximize the angular margins between rival classes, and almost all methods are especially proposed for face recognition. To this end, Liu et al. [12, 13] proposed the SphereFace method which uses the angular softmax (A-softmax) loss that enables to learn angularly discriminative features. Zhao et al. [14] proposed the RegularFace method in which A-softmax term is combined with an exclusive regularization term to maximize the between-class separation. Wang et al. [15] introduced the CosFace method which imposes an additive angular margin on the learned features. To this end, they normalize both the features and the learned weight vectors to remove radial variations and then introduce an additive margin term, $m$, to maximize the decision margin in the angular space. A similar method called ArcFace is introduced in [16], where an additive angular margin is added to the target angle to maximize the separation in angular space. Liu et al. [17] proposed AdaptiveFace method that enables to adjust the margins for different classes adaptively. [18] introduced uniform loss function to learn equidistributed representations for face recognition. We would like to point out that almost all methods that maximize the margin in the angular space are proposed for face recognition. As indicated in [7], these methods work well for face recognition since face class samples in specific classes can be approximated by using linear/affine spaces, and the similarities can be measured well by using the angles between sample vectors in such cases. Linear subspace approximation will work as long as the number of the features is much larger than the number of class specific samples which holds for many face recognition problems. However, for many general classification problems, the training set size is much larger compared to the dimensionality of the learned features and therefore these methods cannot be generalized to the classification applications other than face recognition. In addition to this problem, these methods are also complex since they

have many parameters that must be set by the user as in the methods that maximize the margin in the Euclidean spaces.

**Contributions:** The methods that maximize the margin in Euclidean or angular spaces mentioned above have the shortcomings in the ways that the objective loss functions include many terms that need to be weighted, the class acceptance regions are not compact, or they need additional hard-mining algorithms. In this study, we propose a simple yet effective method that does not have these limitations. Our proposed method maximizes the margin in both the Euclidean and angular spaces. To the best of our knowledge, our proposed method is the first method that maximizes the margin in both spaces. To accomplish this goal, we train a deep neural network that enforces the samples to gather in the vicinity of the class-specific centers that lie on the boundary of a hypersphere. Each class is represented with a single center and the distances between the class centers are equivalent. This corresponds to selection of class centers from the vertices of a regular simplex inscribed in a hypersphere. Both the Euclidean distances and angular distances between class centers are equivalent to each other.

Our proposed method has many advantages over other margin maximizing deep neural network classifiers. These advantages can be summarized as follows:

- The proposed loss function does not have any hyperparameter that must be fixed for classical classification problems, therefore it is extremely easy for the users. For open set recognition, the user has to set two parameters if the background class samples are used for learning.
- The proposed method returns compact and interpretable acceptance regions for each class, thus it is very suitable for open set recognition problems.
- The distances between the samples and their corresponding centers are minimized independently of each other, thus the proposed method also works well for unbalanced datasets.

In contrast, there is only one limitation of the proposed method: The dimension of the CNN features must be larger than or equal to the total number of classes minus 1. To overcome this limitation, we introduced Dimension Augmentation Module (DAM) as explained below.

# 2 Method

## 2.1 Motivation

In this study, we propose a simple yet effective deep neural network classifier that maximizes the margin in both Euclidean and angular spaces. To this end, we introduce a novel classification loss function that enforces the samples to compactly cluster around the class-specific centers that are selected from the outer boundaries of a hypersphere. The Euclidean distances and angles between the centers are equivalent. This is illustrated in Fig. 1. In this figure, the centers representing the classes are denoted by the star symbols whereas the class samples are represented with circles having different colors based on the class memberships. As seen in the figure, all pair-wise distances between the class centers are equivalent, and class centers are located on the boundary of a hypersphere. Moreover, if the hypersphere center is set to the origin, then the angles between the class centers are also same, and the lengths of the centers are equivalent, i.e., $\|\mathbf{s}_i\| = u$, ($u$ is the length of the center vectors). After learning stage, if the class samples are compactly clustered around the centers representing them, we can classify the data samples based on the Euclidean or angular distances from the class centers. Both distances yield the same results if the hypersphere center is set to the origin.

At this point, the question of whether enforcing data samples to lie around the simplex vertices is appropriate or not comes to mind. In fact, high-dimensional spaces are quite different than the low dimensional spaces, and there are many studies showing that the data samples lie on the boundary of a hypersphere when the feature dimensionality, $d$, is high and the number of samples, $n$, is small. For example, Jimenez and Landgrebe [19] theoretically show that the high-dimensional spaces are mostly empty and data concentrate on the outside of a shell (on the outer boundary of a hypersphere). They also show that as the number of dimensions increases, the shell increases its distance from the origin. More precisely, the data samples lie near the outer surface of a growing hypersphere in high-dimensional spaces. In a more recent study, Hall et al. explicitly [20] show that the data samples lie at the vertices of a regular simplex in high-dimensional spaces. These two studies are not contradictory and they support each other since we can always inscribe a regular simplex in

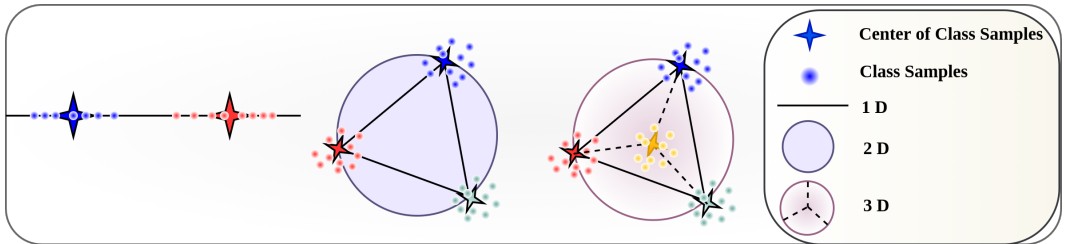

Figure 1: In the proposed method, class samples are enforced to lie closer to the class-specific centers representing them, and the class centers are located on the boundary of a hypersphere. All the distances between the class centers are equivalent, thus there is no need to tune any margin term. The class centers form the vertices of a regular simplex inscribed in a hypersphere. Therefore, to separate $C$ different classes, the dimensionality of the feature space must be at least $C - 1$. The figure on the left shows separation of 2 classes in 1-D space, the middle figure depicts the separation of 3 classes in 2-D space, and the figure on the right illustrates the separation of 4 classes in 3-D space. For all cases, the centers are chosen from a regular $C-$simplex.

145 a hypersphere as seen in Fig. 1. In addition to these studies, [21, 22] show that the eigenvectors
146 of the Laplacian matrices (the matrices computed by operating on similarity matrices in spectral
147 clustering analysis) form a simplex structure, and they use the vertices of resulting simplex for
148 clustering of data samples. In other words, they prove that when the data samples are mapped to
149 Laplacian eigenspace, they concentrate on the vertices of a simplex structure. These studies are also
150 complementary to the studies showing that the high-dimensional data samples lie on the boundary of
151 a growing hypersphere. It is because, as proved in [23], NCuts (Normalized Cuts) [24] clustering
152 algorithm, which is presented as a spectral relaxation of a graph cut problem, maps the data samples
153 onto an infinite-dimensional feature space. Therefore, these data samples naturally concentrate on the
154 vertices of a regular simplex due to the high-dimensionality of the feature space.

155 ## 2.2 Maximizing Margin in Euclidean and Angular Spaces

156 In the proposed method, we map the class samples to compactly cluster around the class centers
157 chosen from the vertices of a regular simplex. All the pair-wise distances between the selected class
158 centers are equivalent. Assume that there are $C$ classes in our data set. In this case, we first need to
159 create a $C$-simplex (some researchers call it $C - 1$ simplex considering the feature dimension, but
160 we will prefer $C$-simplex definition). The vertices of a regular simplex inscribed in a hypersphere
161 with radius 1 can be defined as follows:

$$\mathbf{v}_j = \begin{cases} (C-1)^{-1/2}\mathbf{1}, & j = 1, \\ \kappa\mathbf{1} + \eta\mathbf{e}_{j-1}, & 2 \leq j \leq C, \end{cases} \tag{1}$$

162 where,

$$\kappa = -\frac{1 + \sqrt{C}}{(C-1)^{3/2}}, \eta = \sqrt{\frac{C}{C-1}}. \tag{2}$$

163 Here, $\mathbf{1}$ is an appropriate sized vector whose elements are all 1, $\mathbf{e}_j$ is the natural basis vector in
164 which the $j-$th entry is 1 and all other entries are 0. Such a $C-$simplex is in fact a $C-$dimensional
165 polyhedron where the distances between the vertices are equivalent. It must be noted that the distances
166 between the vertices do not change even if the simplex is rotated or translated. But, the dimension
167 of the feature space must be at least $C - 1$ in order to define such a regular $C-$simplex. Next, we
168 must define the radius, $u$, of the hypersphere. This term is similar to the scaling parameter used in
169 methods such as ArcFace [16], CosFace [15], etc. that maximize the margin in angular spaces. As
170 the dimension increases, it must also increase since the studies [19] show that the hypersphere whose
171 outer shells include the data also grows as the dimension is increased. We set $u = 64$ as in ArcFace
172 method. Then, we set the class centers that will represent the classes as,

$$\mathbf{s}_j = u\mathbf{v}_j, \quad j = 1, ..., C. \tag{3}$$

173 The order of selection of centers does not matter since the distances among all centers are equivalent.
174 Now, let us consider that the deep neural network features of training samples are given in the form

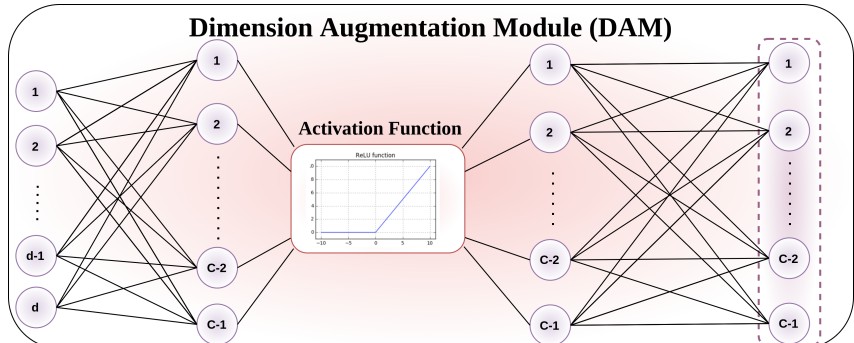

Figure 2: The plug and play module that will be used for increasing feature dimension. It maps $d-$dimensional feature vectors onto a much higher $(C-1)-$ dimensional space.

$(\mathbf{f}_i, y_i)$, $i = 1, \ldots, n$, $\mathbf{f}_i \in \mathbb{R}^d$, $y_i \in \{j\}$ where $j = 1, ..., C$. Here, $C$ is the total number of known classes, and we assume that the feature dimension $d$ is larger than or equal to $C - 1$, i.e., $d \geq C - 1$. In this case, the loss function of the proposed method can be written as,

$$\mathcal{L} = \frac{1}{n} \sum_{i=1}^{n} \|\mathbf{f}_i - \mathbf{s}_{y_i}\|^2 . \tag{4}$$

The loss function includes a single term that aims to minimize the within-class variations by minimizing the distances between the samples and their corresponding class centers which are set to the vertices of a regular simplex. There is no need another loss term for the between-class separation since the selected centers have the maximum possible Euclidean and angular distances among them. As a result, there is no hyperparameter that must be fixed, and the proposed method is extremely easy for the users. Moreover, the data samples compactly cluster around their class centers, therefore the proposed method returns compact acceptance regions for classes, which is crucial for the success of the open set recognition. We call the resulting methods as *Deep Simplex Classifier (DSC)*.

### 2.3 Including Background Class for Open Set Recognition

In open set recognition problems, novel classes (ones not seen during training) may occur at test time, and the goal is to classify the known class samples correctly while rejecting the unknown class samples [25]. Earlier open set recognition methods only used the known class samples during training. However, more recent studies [26, 27, 28] revealed that using the background dataset that includes the samples that come from the classes that are different from the known classes greatly improves the accuracies. Let us represent the deep neural network features of the background samples by $\mathbf{f}_k \in \mathbb{R}^d$, $k = 1, ..., K$. In order to incorporate the background samples, we add an additional loss term that pushes the background samples away from the known class centers as follows:

$$\mathcal{L} = \frac{1}{n} \sum_{i=1}^{n} \|\mathbf{f}_i - \mathbf{s}_{y_i}\|^2 + \lambda \sum_{i=1}^{n} \sum_{k=1}^{K} \max\left(0, m + \|\mathbf{f}_i - \mathbf{s}_{y_i}\|^2 - \|\mathbf{f}_k - \mathbf{s}_{y_i}\|^2\right), \tag{5}$$

where $m$ is the selected threshold, and $\lambda$ is the weighting term. The second loss term enforces the distances between the known class samples and their corresponding class centers to be smaller than the distances between the background class samples and the known class centers by at least a selected margin, $m$. In contrast to our first proposed loss function, this loss function includes two terms that must be set by the users. But, this is necessary only if we use the background class samples.

### 2.4 Dimension Augmentation Module (DAM)

The major limitation of the proposed method is the restriction that the dimension of the feature space must be larger than or equal to $C - 1$, i.e., $d \geq C - 1$. The typical feature dimension size returned by the classical deep neural network classifiers is 2048 or 4096. In this case, the number of classes in our training set cannot exceed 2049 or 4097. However, the number of classes can be larger than these

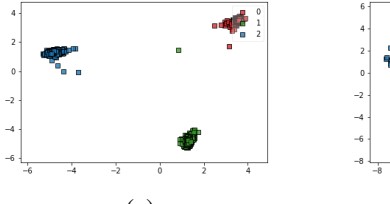 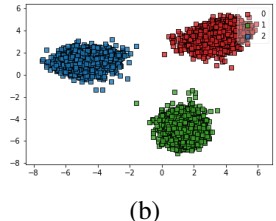 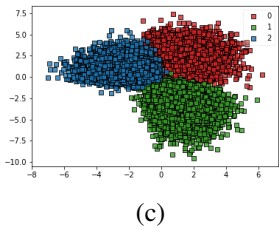

|  (a)  |  (b)  |  (c)  |

Figure 3: Learned feature representations of image samples: (a) the embeddings returned by the proposed method trained with the default loss function given in (4), (b) the embeddings returned by the proposed method trained with the hinge loss, (c) the embeddings returned by the proposed method trained with the softmax loss function.

205  values for some classification tasks, and we cannot use the proposed method in such cases. There are
206  basically two procedures to solve this problem. As a first solution, we can use a method similar to
207  [29] that returns more centers where the distances between centers are approximately equivalent. In
208  this case, the number of centers is increased to $2d + 4$ for $d-$dimensional feature spaces. As a second
209  and a more complete solution, we introduce a module called Dimension Augmentation Module
210  (DAM) that increases the feature dimension size to any desired value. The module is visualized in
211  Fig. 2, and it includes two fully connected layers supported with activation functions. The first fully
212  connected layer maps the $d-$dimensional feature space onto a higher $C - 1$ dimensional space. Then,
213  we apply ReLU (Rectified Linear Unit) activation functions followed by the second fully connected
214  layer. This is similar to kernel mapping idea used in kernel methods [30, 31] in the spirit with the
215  exception that we explicitly map the data to higher dimensional feature space as in [32, 33].

## 3  Experiments

### 3.1  Illustrations and Ablation Studies

218  Here, we first conducted some experiments to visualize the embedding spaces returned by the various
219  loss functions using the vertices of the regular simplex. For this illustration experiment, we designed
220  a deep neural network where the output of the last hidden layer is set to 2 for visualizing the learned
221  features. As training data, we selected 3 classes from the Cifar-10 dataset. We would like to point out
222  that we can use different loss functions in addition to our default loss function given in (4) once we
223  determine the vertices of the simplex that will represent the classes. To this end, we used two other
224  loss functions: The first one is the hinge loss that minimizes the distances between the samples and
225  their corresponding class center if the distance is larger than a selected threshold,

$$\mathcal{L}_{hinge} = \frac{1}{n} \sum_{i=1}^{n} \max \left( 0, \|\mathbf{f}_i - \mathbf{s}_{y_i}\|^2 - m \right). \tag{6}$$

226  This loss function does not minimize the distances between the samples and their corresponding
227  centers if the distances are already smaller than the selected threshold, $m$. This way class-specific
228  samples are collected in a hypersphere with radius, $m$. For the second loss function, we used the
229  variant of the softmax loss function where the weights are fixed to the simplex vertices as in,

$$\mathcal{L}_{softmax} = -\frac{1}{n} \sum_{i=1}^{n} \log \frac{e^{\mathbf{s}_{y_i}^\top \mathbf{f}_i + b_{y_i}}}{\sum_{j=1}^{C} e^{\mathbf{s}_j^\top \mathbf{f}_i + b_j}} \tag{7}$$

230  For the softmax loss, we fix the classifier weights to the pre-defined class centers and we only update
231  features of the samples by using back-propagation. We set the hypersphere radius to, $u = 5$, since
232  this is a simple dataset.

233  The embeddings returned by the deep neural networks using different loss functions are plotted in
234  Fig. 3. The first figure on the left is obtained by our default loss function that does not need any
235  parameter selection. All data samples are compactly clustered around their class means as expected.
236  The second loss function using the hinge loss returns spherical distributions based on the selected

margin, $m$, and the classes are still separable by a margin. In contrast, when the softmax is used with the simplex vertices, the data samples are very close and they overlap since there is no margin among the classes. Therefore, our default loss function seems to be the best choice among all tested variants since it does not need fixing any parameter and returns compact class regions.

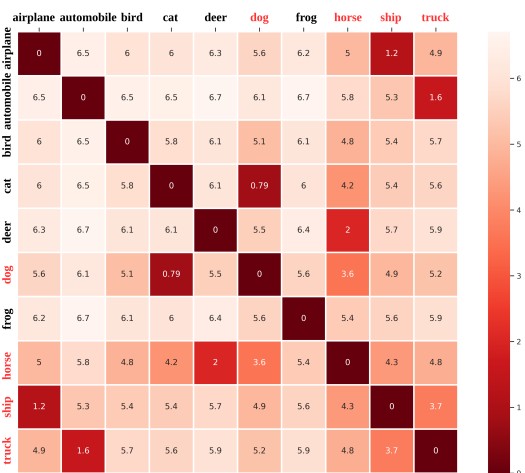

Figure 4: The distance matrix computed by using the centers of the testing classes. The four classes that are not used in training are closer to their semantically related classes in the learned embedding space.

We also conducted experiments to see if the proposed method returns meaningful feature embeddings where the semantically and visually similar classes lie close to each other in open set recognition settings. It should be noted that the semantic relationships are not preserved for the training classes since the Euclidean and angular distances between the class centers are equivalent. However, if the proposed method returns good CNN features, we expect the samples belonging to classes not used in training to lie closer to their semantically related training classes. To verify this, we trained our proposed method by using 6 classes from the Cifar-10 dataset: airplane, automobile, bird, cat, deer, and frog. Then, we extracted the CNN features of all testing data coming from 10 classes by using the trained network. Then, we computed the average CNN feature vector of each class, and computed the distances between them. Fig. 4 illustrates the computed distances between the centers. The distances between the classes used for training are similar and they change between 5.8 and 6.7. The four classes, the dog, horse, ship, and truck classes, that are not used for training are represented with red color in the figure. As seen in the figure, the dog class is closest to its semantically similar cat class, the truck class is closer to its semantically similar automobile class, the horse class is closest to the deer class, and the ship class is closer to the visually similar airplane class (since the backgrounds - blue sky and sea - are mostly similar for these two classes). This clearly shows that the proposed method returns semantically meaningful embeddings.

## 3.2 Open Set Recognition Experiments

For open set recognition, we need to split the datasets into *known* and *unknown* classes. To this end, we used the common standard settings that are also applied for testing other recent open set recognition methods. The details of each dataset and its open set recognition setting are given below. By following the standard protocol, random splitting of each dataset into known and unknown classes is repeated 5 times, and the final accuracies are averages of the results obtained in each trial.

We compared our proposed method, Deep Simplex Classifier (DSC), to other state-of-the-art open set recognition methods including Softmax, OpenMax [25], C2AE [34], CAC [27], CPN [35], OSRCI [36], CROSR [37], RPL [38], Objecttosphere [39], and Generative-Discriminative Feature Representations (GDFRs) [40] methods. We used the same network architecture used in [36] as our backbone network for all datasets with the exception of TinyImageNet dataset, where we preferred a deeper Resnet-50 architecture for this dataset. We started the training from completely random weights (without any fine-tuning). Therefore, our proposed method is directly comparable to the published results in [36] for majority of the tested datasets.

Table 1: AUC Scores (%) of open set recognition methods on tested datasets ($n.r.$ stands for not reported).

| Methods | Mnist | Cifar10 | SVHN | Cifar+10 | Cifar+50 | TinyImageNet |
|---|---|---|---|---|---|---|
| DSC (Ours) | **99.6** $\pm$ 0.1 | 93.8 $\pm$ 0.3 | 95.3 $\pm$ 0.8 | **99.1** $\pm$ 0.2 | **98.4** $\pm$ 0.3 | **82.5** $\pm$ 1.8 |
| Softmax | 97.8 $\pm$ 0.2 | 67.7 $\pm$ 3.2 | 88.6 $\pm$ 0.6 | 81.6 $\pm$ $n.r.$ | 80.5 $\pm$ $\pm n.r.$ | 57.7 $\pm$ $n.r.$ |
| OpenMax | 98.1 $\pm$ 0.2 | 69.5 $\pm$ 3.2 | 89.4 $\pm$ 0.8 | 81.7 $\pm$ $n.r.$ | 79.6 $\pm$ $n.r.$ | 57.6 $\pm$ $n.r.$ |
| G-OpenMax | 98.4 $\pm$ 0.1 | 67.5 $\pm$ 3.5 | 89.6 $\pm$ 0.6 | 82.7 $\pm$ $n.r.$ | 81.9 $\pm$ $n.r.$ | 58.0 $\pm$ $n.r.$ |
| C2AE | 98.9 $\pm$ 0.2 | 89.5 $\pm$ 0.9 | 92.2 $\pm$ 0.9 | 95.5 $\pm$ 0.6 | 93.7 $\pm$ 0.4 | 74.8 $\pm$ 0.5 |
| CAC | 99.1 $\pm$ 0.5 | 80.1 $\pm$ 3.0 | 94.1 $\pm$ 0.7 | 87.7 $\pm$ 1.2 | 87.0 $\pm$ 0.0 | 76.0 $\pm$ 1.5 |
| CPN | 99.0 $\pm$ 0.2 | 82.8 $\pm$ 2.1 | 92.6 $\pm$ 0.6 | 88.1 $\pm$ $n.r.$ | 87.9 $\pm$ $n.r.$ | 63.9 $\pm$ $n.r.$ |
| OSRCI | 98.8 $\pm$ 0.1 | 69.9 $\pm$ 2.9 | 91.0 $\pm$ 0.6 | 83.8 $\pm$ $n.r.$ | 82.7 $\pm$ $-$ | 58.6 $\pm$ $n.r.$ |
| CROSR | 99.1 $\pm$ $n.r.$ | 88.3 $\pm$ $n.r.$ | 89.9 $\pm$ $n.r.$ | 91.2 $\pm$ $n.r.$ | 90.5 $\pm$ $n.r.$ | 58.9 $\pm$ $n.r.$ |
| RPL | 98.9 $\pm$ 0.1 | 82.7 $\pm$ 1.4 | 93.4 $\pm$ 0.5 | 84.2 $\pm$ 1.0 | 83.2 $\pm$ 0.7 | 68.8 $\pm$ 1.4 |
| GDFRs | $n.r.$ | 83.1 $\pm$ 3.9 | **95.5** $\pm$ 1.8 | 92.8 $\pm$ 0.2 | 92.6 $\pm$ 0.0 | 64.7 $\pm$ 1.2 |
| Objecttosphere | $n.r.$ | **94.2** $\pm$ $n.r.$ | 91.4 $\pm$ $n.r.$ | 94.5 $\pm$ $n.r.$ | 94.4 $\pm$ $n.r.$ | 75.5 $\pm$ $n.r.$ |

### 3.2.1 Datasets

**Mnist, Cifar10, SVHN:** By using the standard setting, Mnist, Cifar10, and SVHN datasets are split randomly into 6 known and 4 unknown classes. We used 80 Million Tiny Images dataset [41] as the background class.

**Cifar+10, Cifar+50:** For Cifar+$N$ experiments, we use 4 randomly selected classes from Cifar10 dataset for training, and $N$ non-overlapping classes chosen from Cifar100 dataset are used as unknown classes as in [35, 27, 37, 38]. We used 80 Million Tiny Images dataset [41] as the background class.

**TinyImageNet:** For TinyImageNet [42] experiments, we randomly selected 20 classes as known classes and 180 classes as unknown classes by following the standard setting. We used 80 Million Tiny Images dataset [41] as the background class.

### 3.2.2 Results

For open set recognition, Area Under the ROC curve (AUC) scores are used for measuring the detection of performance of the unknown samples. In addition, we also report the closed-set accuracy for measuring the classification performance on known data by ignoring the unknown samples as in [35, 36] (these results are given in Appendix). AUC scores are given in Table 1. As seen in the table, our proposed method achieves the best accuracies on all datasets with the exception of Cifar 10 and SVHN datasets. The performance difference is very significant especially on Cifar+10, Cifar+50 and TinyImageNet datasets.

## 3.3 Closed Set Recognition Experiments

### 3.3.1 Experiments on Moderate Sized Datasets

Here, we conducted closed set recognition experiments on moderate sized datasets. Our proposed method did not need DAM since the feature dimension is much larger than the number of classes in the training set for these experiments. We compared our results to the methods that maximize the margin in Euclidean or angular spaces. We implemented the compared methods by using provided source codes by their authors, and we used the ResNet-18 architecture [43] as backbone for all tested methods. Therefore, our results are directly comparable.

Table 2: Classification accuracies (%) on moderate sized datasets.

| Methods | Mnist | Cifar-10 | Cifar-100 |
|---|---|---|---|
| DSC (Ours) | **99.7** | **95.9** | **79.5** |
| Softmax | 99.4 | 94.4 | 75.3 |
| Center Loss | **99.7** | 94.2 | 76.1 |
| ArcFace | **99.7** | 94.8 | 75.7 |
| CosFace | **99.7** | 95.0 | 75.8 |
| SphereFace | **99.7** | 94.7 | 75.1 |

Classification accuracies are given in Table 2. For Mnist datasets, majority of the tested methods yield the same accuracy, but our proposed DSC method outperforms all tested methods on the Cifar-10 and Cifar-100 datasets. The performance difference is significant especially on the Cifar-100 dataset. These results verify the superiority of the margin maximization in both Euclidean and angular spaces. Achieving the best accuracies is encouraging, because our proposed method is very simple and does not need any parameter tuning, yet it outperforms more complex methods.

### 3.3.2 Experiments on Large-Scale Datasets

For all face verification tests, we used the same network trained on large-scale face dataset by following the standard setting. To this end, we trained the proposed classifier on MS1MV2 dataset [16], which is a cleaned version of MS-Celeb-1M dataset [44]. This dataset includes approximately 85.7K individuals. We removed the classes including less than 100 samples, which left us approximately 18.6K individuals for training. The number of classes is much larger than the feature dimension, $d = 2048$, thus we used DAM to increase the CNN feature dimension. The ResNet-101 architecture is used as backbone. Once the network is trained, we used the resulting architecture to extract deep CNN features of the face images coming from the test datasets.

As test datasets, we used Labeled Faces in the Wild (LFW) [45], Cross-Age LFW (CALFW) [46], Cross-Pose LFW (CPLFW) [47], Celebrities in Frontal-Profile data set (CFP-FP) [48] and AgeDB [48]. We evaluated the proposed methods by following the standard protocol of unrestricted with labeled outside data [45], and report the results by using 6,000 pair testing images on LFW, CALFW, CPLFW, and AgeDB. However, 7,000 pairs of testing images are used for CFP-FP by following the standard setting. The results are given in Table 3. As seen in the results, the proposed method using DAM outperforms the classifiers using softmax and Center loss, but accuracies are lower than the recent state-of-the-art methods. These results indicate that the DAM solves the dimension problem partially, but it must be revised for obtaining better accuracies.

Table 3: Verification rates (%) on different datasets.

| Method | LFW | CALFW | CPLFW | CFP | AgeDB |
|---|---|---|---|---|---|
| DSC | 99.6 | 91.3 | 90.3 | 94.3 | 96.0 |
| VGGFace2 | 99.4 | 90.6 | 84.0 | —— | —— |
| Center Loss | 99.3 | 85.5 | 77.5 | —— | —— |
| ArcFace (ResNet-101) | **99.8** | **95.5** | **92.1** | 95.6 | —— |
| CosFace | 99.7 | 93.3 | **92.1** | —— | **97.7** |
| SphereFace | 99.4 | 93.3 | **92.1** | 94.4 | **97.7** |

## 4 Summary and Conclusion

In this paper, we proposed a simple and effective deep neural network classifier that maximizes the margin in both the Euclidean and angular spaces. The proposed method returns embeddings where the class-specific samples lie in the vicinity of the class centers chosen from the vertices of a regular simplex. The proposed method is very simple in the sense that there is no parameter that must be fixed for classical closed set recognition settings. Despite its simplicity, the proposed method achieves the state-of-the-art accuracies on open set recognition problems since the samples of unknown classes are easily rejected by using the distances from the class-specific centers. Moreover, our proposed method also outperformed other state-of-the-art classification methods on closed set recognition setting when moderate sized datasets are used. The proposed method has a limitation regarding learning in large-scale datasets. We introduced DAM in order to solve this problem. Although DAM partially solved the existing problem, we could not get state-of-the-art accuracies on large-scale face recognition problems. As a future work, we are planning to improve DAM by changing its architecture and activation functions.

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

# A  Appendix

Here, we first explain the implementation details of the proposed deep neural network classifier, and give the parameters used for the utilized deep neural network classifier architecture. Then, we reported the closed-set accuracies of tested methods on open set recognition datasets.

## A.1  Implementation Details

For open set recognition, we used the same network architecture used in [36] as our backbone network for all datasets with the exception of TinyImageNet dataset, where we preferred a deeper Resnet-50 architecture for this dataset. The learning rate is set to $0.1$. For open set recognition experiments, we set $\lambda = \frac{1}{2 \times batch\_size^2}$, and $m = u/2$, where $u$ is the hypersphere radius.

We do not need these parameters for closed set recognition. For closed-set recognition experiments, we used the ResNet-18 architecture as backbone for moderate sized datasets, and the ResNet-101 architecture is used for large-scale face recognition dataset. For updating network weights, we used Adam optimization strategy for large-scale face recognition whereas SGD (stochastic gradient descent) is used for moderate size datasets. The learning rate is set to $10^{-3}$ for face recognition and to $0.5$ for moderate sized datasets.

## A.2  Closed-Set Accuracies on Open Set Recognition Datasets

Closed-set accuracies of the open-set recognition methods are given in Table 4. Our proposed method also obtains the best closed-set accuracies among the tested methods with the exception of SVHN

dataset. This clearly shows that the proposed method is very successful both at the rejection of the unknown samples and classification of the known samples correctly.

Table 4: Closed-Set accuracies (%) of open set recognition methods on tested datasets.

| Methods | Mnist | Cifar10 | SVHN | Cifar+10 | Cifar+50 | TinyImageNet |
|---------|-------|---------|------|----------|----------|--------------|
| DSC (Ours) | $\mathbf{99.8} \pm 0.1$ | $\mathbf{96.1} \pm 1.4$ | $96.5 \pm 0.3$ | $\mathbf{97.6} \pm 0.5$ | $\mathbf{97.9} \pm 0.5$ | $\mathbf{83.3} \pm 2.2$ |
| Softmax | $99.5 \pm 0.2$ | $80.1 \pm 3.2$ | $94.7 \pm 0.6$ | $n.r.$ | $n.r.$ | $n.r.$ |
| OpenMax | $99.5 \pm 0.2$ | $80.1 \pm 3.2$ | $94.7 \pm 0.6$ | $n.r.$ | $n.r.$ | $n.r.$ |
| G-OpenMax | $99.6 \pm 0.1$ | $81.6 \pm 3.5$ | $94.8 \pm 0.8$ | $n.r.$ | $n.r.$ | $n.r.$ |
| CPN | $99.7 \pm 0.1$ | $92.9 \pm 1.2$ | $\mathbf{96.7} \pm 0.4$ | $n.r.$ | $n.r.$ | $n.r.$ |
| OSRCI | $99.6 \pm 0.1$ | $82.1 \pm 2.9$ | $95.1 \pm 0.6$ | $n.r.$ | $n.r.$ | $n.r.$ |
| CROSR | $99.2 \pm 0.1$ | $93.0 \pm 2.5$ | $94.5 \pm 0.5$ | $n.r.$ | $n.r.$ | $n.r.$ |

