# OpenReview forum: "Reaching Nirvana: Maximizing the Margin in Both Euclidean and Angular Spaces for Deep Neural Network Classification"
_NeurIPS.cc/2022/Conference — NeurIPS 2022 Submitted_

### Official Review · Reviewer_mqvu · 2022-07-12

**Rating:** 4
**Confidence:** 4
**Soundness:** 2 fair
**Presentation:** 2 fair
**Contribution:** 2 fair

**Summary:**

This paper introduces a Deep Simplex Classiﬁer (DSC) that maximizes the inter-class margins in both Euclidean and angular spaces for the open set recognition problem. Specifically, this method regards the vertices of a regular C-simplex in the (C-1)-dimensional feature space as the class centers of known C classes in the dataset and then encourages the class samples to be as close to their corresponding centers as possible. In cases that the feature dimension is smaller than C-1, a Dimension Augmentation Module (DAM) is further proposed to expand the feature dimension to C-1 to make it possible to construct the C-simplex. The open set and closed set recognition experiments are conducted on Mnist, Cifar10, SVHN, Cifar100, and Tiny ImageNet datasets while the face recognition experiments are conducted on MS1Mv2 dataset to show the effectiveness of the proposed method.

**Questions:**

1. The novelty of the proposed method needs further clarification. More discussions and empirical comparisons between the proposed method and [1], [2], [3] are needed since they are quite related in methodology or goal. As the relations between each pair of classes are different, is it necessary to make the pairwise distances between any two centers identical as in DSC or what is its key advantage?
2. The motivation should be expressed more clearly. The authors mostly emphasize the complexity of hyperparameters of existing methods in introduction while do not explain how would two kinds of distances complement each other and what is the key advantage of considering both spaces in open set recognition? Besides, the presentation of introduction could be further improved.
3. How to better alleviate the feature dimension restriction for the proposed method without introducing too many parameters, especially when there are a large number of classes as in face recognition?
4. The ablation studies for hyperparameters u, \lambda and m in Equation 5 are needed.
5. For the open set recognition task, the proposed DSC uses a large amount of data from 80 Million Tiny Images dataset as the background class in Section 3.2.1, but the existing methods do not adopt this setting, for which I think the comparison in Table 1 is unfair. In line 269, a deeper network is employed for the Tiny ImageNet dataset, is it the same with the settings in existing works?
6. For the closed set recognition task in Table 2, do authors tune the hyperparameters in SphereFace, CosFace, and ArcFace for current classification datasets or directly follow the settings in original papers?


**Limitations:**

No. The authors do not adequately address the feature dimension restriction of the proposed DSC. The proposed DAM introduces a large number of parameters when the number of classes is large.

**Strengths And Weaknesses:**

Strengths:
1. The practice of regarding the vertices of a regular C-simplex in (C-1)-dimensional space as the C classes’ centers in the training dataset is new in open set recognition task, which has a clear geometric interpretation.
2. The proposed method is clearly explained and easy to follow.
3. The authors conduct the illustration experiment to ease the understanding of the effect of DSC and evaluate the proposed method extensively on open set recognition, closed set recognition, and large-scale face recognition tasks.

Weaknesses:
1. The idea of maximizing the inter-class margins by explicitly manipulating the class centers is not new. For fixed centers, in Section 3.2 of [1], Do et al. propose to regard the basis vectors of the C-dimensional space as the C classes’ centers which are also the vertices of a C-simplex. The only difference is that the feature dimension in [1] is one larger than that in this paper. For learnable centers, Hayat et al. [2] also encourage the pairwise distances between any two class centers to be similar in a learnable way to bring uniformly separated centers in Euclidean space. Besides, Liu et al. [3] propose a minimum hyperspherical energy (MHE) regularizer to uniformly distribute the class centers on a hypersphere for maximizing the inter-class separability. These two methods share similar goals with Deep Simplex Classiﬁer (DSC) but do not require that the feature dimension is larger than C-2. As the relations between each pair of classes are different, is it necessary to make the pairwise distances between any two centers identical as in DSC or what is its advantage? I think more discussions and empirical comparisons are needed considering these strongly related existing works.
2. The motivation of this paper is not clearly expressed yet. The authors claim that the Euclidean and Cosine distances would complement each other in abstract and finally choose to maximize the margin in both the Euclidean and angular spaces in introduction. However, the authors mostly emphasize the complexity of hyperparameters of existing methods in introduction while do not explain how would two kinds of distances complement each other and what is the key advantage of considering both spaces in open set recognition? Besides, the authors simply list and explain a series of existing works one by one in introduction with few summaries, which is kind of redundant, lacks logical coherence, and could be further improved.
3. The proposed DSC is strictly restricted by the requirement that the feature dimension is not less than C-1. The Dimension Augmentation Module (DAM) would only alleviate this issue when C is small while with a large C, DAM will introduce a large number of parameters. In the face recognition experiment, with 18.6K identities, the number of parameters of the last linear layer in DAM is about 18.6K*18.6K \approx 346M, which is seven times more than the number of parameters of the ResNet-101 backbone (45M) and hinders the practical usage. I think a better solution is needed and essential for the general usage of DSC.
4. The authors claim that DSC does not have any hyperparameters but actually there is a hyperparameter u denoting the radius of the hypersphere. The authors simply set it to 64 following the practice in face recognition, which lacks theoretical and empirical guarantee as this paper does not focus on the task of face recognition and uses datasets beyond face recognition datasets. For this reason, an ablation study is needed to investigate the effect of the value of u. Moreover, the ablation study for the hyperparameters \lambda and m in Equation 5 in open set recognition is also needed. Besides, I also have some concerns about the experiments. First, for the open set recognition task, the proposed DSC uses a large amount of data from 80 Million Tiny Images dataset as the background class in Section 3.2.1, but the existing methods do not adopt this setting, for which I think the comparison in Table 1 is unfair. In line 269, a deeper network is employed for the Tiny ImageNet dataset, is it the same with the settings in existing works? Second, for the closed set recognition task in Table 2, do authors tune the hyperparameters in SphereFace, CosFace, and ArcFace for current classification datasets or directly follow the settings in original papers? Besides, the methods mentioned in the first comment should also be compared in the experiments since they are quite related with the proposed method.

[1] Do, Thanh-Toan, et al. "A theoretically sound upper bound on the triplet loss for improving the efficiency of deep distance metric learning." In CVPR. 2019.
[2] Hayat, Munawar, et al. "Gaussian affinity for max-margin class imbalanced learning." In ICCV. 2019.
[3] Liu, Weiyang, et al. "Learning towards minimum hyperspherical energy." In NeurIPS. 2018.

---

> ### Author Response · Authors · 2022-08-01
> **Response for Reviewer mqvu**
>
> 1) We thank the reviewer for pointing out [1] which uses a similar idea as in the proposed method. This paper focuses on distance metric learning, and it uses class centers chosen as the basis vectors of C-dimensional space as anchors. However, there are 2 critical mistakes: The first mistake is to choose the centers from the surface of a unit hypersphere. As discussed in our paper, the data samples lie near the surface of a growing hypersphere as the dimension increases. Therefore, setting the hypersphere radius to 1 is not suitable for most problems, and similar findings and discussions are given in ArcFace paper. The second mistake is to use a fully connected layer alone for increasing the dimensionality. A fully connected layer just uses the linear combination of existing features and the resulting space has the same dimensionality (they need to use activation functions to introduce nonlinearity). Therefore, this method will not work for large-scale problems. Moreover, our proposed method is much simpler and run-time complexity of the proposed method is significantly less. For empirical comparison, we will run this method on the same datasets we used and report the results in the final version.
> Regarding the paper [2], the method is similar to Uniformface which proposes a classification loss function for learning uniformly distributed representations on the hypersphere. These methods are complex since they need many hyperparameters such as different weights for loss terms and margin terms, and provide approximate solutions.  On the other hand, our proposed method uses the result of the optimal solution given in Equations (1-2) in our paper. Also, enforcing the distances between pairwise class centers to have the same value eliminates the need for introducing margin terms and yields a much simpler method. Setting margin terms is a difficult problem especially for deep learning methods since the feature representations also change during training.
> For comparison between our proposed method and [3], see our reply for Reviewer 2 (a4BR).
> 2) Regarding the importance of maximizing the margin in both the angular and Euclidean spaces, this helps to apply the proposed method to various object classification problems. As stated on page 2, the methods maximizing the margin in angular spaces are used only for face recognition problems, where the classes can be approximated with linear/affine subspaces (in this case ArcFace and similar methods estimate the most discriminative directions spanning the subspaces as the class-specific weights and use them for classification). However, this approximation does not work well for more general object classification problems such as Cifar100 or ImageNet. For such problems, methods that minimize the within-class variances and maximize the inter-class separation based on Euclidean distances work better. As a result, maximizing the margin in both spaces makes our proposed method well-suited to any kind of classification problem such as face recognition or more general object classification problems. We will make it more clear in the final version.
> 3)  We can provide a better solution by designing our own architecture instead of using our proposed plug and play DAM module. To this end, we can avoid the fully connected layers that are used for dimension reduction in the last layers of deep CNNs. For example, in ResNet architectures, the dimension of the feature space is 25088 just before fully connected layers. Using 25088 dimensional feature space is enough for training the large-scale MS1MV2 dataset without any need for dimension increase. At this point, we would like to point out the fact that all methods ([1] and [3] ) that target uniformly distributed class centers on the hypersphere have the same dimension problem. The authors simply did not realize it or they did not conduct experiments on large-scale as we did.
> 4) We already conducted experiments by selecting different values for u and \lambda parameters. We will add these results to the Appendix in the final version. Regarding u parameter, experiments verify that the selection of u is not very important as long as it is not fixed to small values such as 1. For larger dimensions, we need higher u values. Also, after some value, increasing u value does not change the results much. Regarding using 80 million Tiny Images, it is used as outlier exposure dataset for anomaly detection problems. For open set recognition, some studies state that they employ background class, but they do not specify the dataset or some studies use GANs to create background samples. Therefore, we decided to use 80 million Tiny Images as background class as in anomaly detection problems. Regarding using a deeper network for Tiny ImageNet dataset, some studies prefer deeper networks as in our study. For example, [38] employs a deeper network Wide-ResNet-40-4 for better accuracies. For all compared methods, we tried to tune the hyperparameters.

---

> > ### Author Response · Authors · 2022-08-03
> > **additional details related to the 1st response**
> >
> > Here, we would like to add additional comments since we could not fit some details in our first response.
> >
> > Regarding comparison to [1], although this paper focuses on distance metric learning, it uses class centers chosen as the basis vectors of C-dimensional space as anchors. Then, as in triplet loss, it attempts to minimize the distances between the data samples and the corresponding class centers (anchors) and to maximize the distances between the samples and rival class centers. Therefore, this method is quite different form our proposed one, and it can be seen as a quantized distance metric learning approach, where the anchors are set to some fixed centers.  However, the authors make 2 critical mistakes: The first mistake is to choose the centers from the surface of a unit hypersphere (a hypersphere with radius 1). As we discussed in our paper, the data samples lie near the surface of a growing hypersphere as the dimension increases. Therefore, setting the hypersphere radius to 1 is wrong, and similar findings and discussions are given in ArcFace [16] and CosFace [15] papers. The second mistake is to use a fully connected layer alone for increasing the dimensionality. A fully connected layer just uses the linear combination of existing features and the resulting space has the same dimensionality. Therefore, this method will not work for large-scale problems. They have to use activation functions to introduce nonlinearity and increase the dimension. If we theoretically compare this method to ours, our proposed method is much simpler and run-time complexity of the proposed method is significantly less. For empirical comparison, we will run this method on the same datasets (with the exception of large-scale face recognition datasets) we used in our paper and report the results in the final version.
> >
> > Regarding comparison to the paper [2], the method is similar to Uniformface which proposes a classification loss function for learning uniformly distributed representations on the hypersphere. There are such approaches in the literature, but they are complex since they need many hyperparameters such as different weights for loss terms and margin terms. In contrast, our proposed method does not have such limitations as stated in our paper. In addition, we would like to point out the fact that, all these methods will end up an approximation solution whereas our proposed method uses the result of the optimal solution given in Equations (1-2) in our paper.
> >
> > Lastly, regarding the scale parameter u, we have already conducted experiments by selecting different values. Experiments verify that the selection of u is not very important as long as it is not fixed to small values such as 1. Theoretically, the data samples lie on the surface of a growing hypersphere as the dimension increases. For smaller dimensions, we can choose smaller values of u as we did for illustrations experiments (we fixed u to 5 for 2-dimensional inputs). But, for larger dimensions we need higher values. Also, after some value, increasing u value does not change the results much. These are the accuracies we obtained for Cifar100 dataset for various u values:
> >
> > u=32, accuracy = 76.2%
> >
> > u=64, accuracy = 79.5%
> >
> > u= 100, accuracy = 79.4%
> >
> >  u= 150, accuracy = 79.9%

---

> > > ### Comment · Reviewer_mqvu · 2022-08-10
> > > **Reply to the response**
> > >
> > > Thanks for the response. The authors have addressed some of my concerns, but I still think the weaknesses of this paper outweigh the strengths.
> > > 1. The motivation of this paper is still not clear enough. The authors claim that they maximize the margins in both the angular and Euclidean spaces to make the proposed method suitable to any kinds of classification problems. However, this cannot explain why the proposed method works better than those who only optimize the Euclidean space in general object classification problems. Besides, the proposed method doesn’t show superiority in face recognition as in Table 3.
> > > 2. Considering the problem of DAM, the authors provide a new solution to flatten the final feature map of ResNet to get a 25088-d feature vector to avoid a large number of parameters in DAM. However, the authors haven’t evaluated this solution empirically and the number of classes are still restricted by the spatial size and the number of channels of the final feature map. Besides, the methods in [1] and [3] don’t have severe dimension problems as the proposed method since the number of parameters of the final linear layer in [1] is only 256 \times C and [3] doesn’t project the dimension of feature vectors to C.
> > > 3. The comparisons are unfair as the proposed DSC uses a large amount of data from 80 Million Tiny Images dataset as the background class. Besides, a deeper network is employed for the Tiny ImageNet dataset. These settings are inconsistent with the other methods, leading to unfair comparisons. When the settings of background class of some existing methods are unknown, I think reimplementing the methods with 80 Million Tiny Images dataset as the background class is a fair comparison approach. In addition, it is unknown how to set the hyperparameter m in Equation 5.

---

> > > > ### Author Response · Authors · 2022-08-10
> > > > **response for the reviewer**
> > > >
> > > > 1) We have written a Motivation subsection to explain our motivation. There are theoretical proofs showing that the data samples lie on the vertices of a regular simplex (equivalently on the boundary of a hypersphere) in high-dimensional spaces. Therefore, it makes perfect sense to map the class-specific data samples to the centers chosen as the vertices of the simplex. Experimental studies support these claims. Our proposed methods outperform other methods using Euclidean distances only, and the reason for this is simple. None of these methods choses the class specific centers from the vertices of a regular simplex and attempts to minimize the distances between the samples and corresponding centers as in the proposed method. Please read our Motivation subsection for more details.
> > > >
> > > > 2) Regarding our alternative solution to DAM, we just wanted to point out that there are alternative solutions to increase dimensionality if we use our special architectures since the reviewers complained about the complexity of the DAM module. DAM already achieved satisfactory results and we did not need to use that alternative solution. The reviewer is also wrong regarding the paper [1] and it has the same dimensionality problem as in the proposed method. The authors use orthogonal vectors to represent the classes (see 3.2. Centroid Generation in [1]). This is basic linear algebra. If there are C classes to represent, this means that the feature dimension must be at least C in order to get an orthogonal set of vectors. Furthermore, the authors of [1] also accept this restriction and they propose to increase the dimension from D to C by adding a fully connected layer to project features from D dimensions to C dimensions (see Section 3. Discriminative Loss in [1]). However, this is not enough since using linear combinations does not increase the dimension, they have to use nonlinear activation functions as in our proposed DAM module.
> > > > Regarding [3], that method does not have the dimension problem, but the same authors publish a recent paper and mention methods using orthogonal/orthonormal weight vectors for hyperspectral uniformity. All these variants will also have the dimension problem similar to [1] since the dimension D must be larger than or equal to C to represent the C classes with linearly independent orthogonal basis vectors.
> > > >
> > > > 3) We explicitly indicated that there are also methods using larger networks for the Tiny ImageNet dataset as in the proposed method. In addition, open set recognition methods also utilize background class samples by creating samples via GANs. Therefore, we believe that our comparisons are fair. We did not implement open set recognition methods by ourselves, we just reported the accuracies from the literature. Therefore, reimplementing all these methods by using background class is not feasible. But, we can also report our accuracies without using background class.
> > > >
> > > > Regarding m parameter, we did not have any trouble for fixing it since our centers are fixed to certain positions. We already know the distances between the class centers chosen as simplex vertices. All distances are equal, we simply checked the largest intra-class distances within classes and determined a margin based on this. Setting margin term to half of the radius worked well for all cases.

---

### Official Review · Reviewer_a4BR · 2022-07-23

**Rating:** 6
**Confidence:** 4
**Soundness:** 2 fair
**Presentation:** 2 fair
**Contribution:** 2 fair

**Summary:**

The paper studies an interesting problem of fixing a set of equi-distance classifiers and then learning features to the corresponding class center (i.e., a vertice in a simplex). The specific form of the loss function is simple and aims to minimize the Euclidean distance between learned feature and the class center (on a hypersphere). This will automatically guarantee maximum separation of class centers and also lead to discriminative feature representation. The paper uses this method in both closed-set and open-set recognition, and shows some improvements.

**Questions:**

- I am wondering whether the authors only uses the proposed Euclidean loss in closed-set recognition (essentially center loss on a simplex)? It is a bit difficult to believe that it can actually outperform Softmax-based margin losses (say ArcFace). Using only Euclidean losses is actually ineffective in classification problems in my experience (it may work for clustering problems like face verification).

-  In the CIFAR-10 experiment, what is feature dimension in the proposed method? How the simplex is constructed if you are using the same feature dimension as the others? Are the authors using 9-dimension features?

**Strengths And Weaknesses:**

I general, I think this is an interesting idea by taking advantages of the simplex and using it as a set of classifiers to guide the feature learning.

Strengths:

- The core idea is interesting. Connecting the simplex and maximum separated classifiers is interesting. Using it as an inductive bias to guide the feature learning is intuitive and seems to make lot of senses.

- The paper also considers the weakness of this method due to the limitation in dimension. The propose dimension augmentaion module aims to address this limitation (however, it inevitably introduces additional trainable parameters and it might not be a fair comparison to the other baselines any more).

- The paper considers both closed-set and open-set recognition and show some improvements over existing baselines.

Weaknesses:

- As the authors have mentioned, the biggest weakness of this method is its restriction to feature dimension. Although the dimension augmentation module can partially address this, it does not solve the problem. For example, if you have a million-level classification (e.g., many face recognition datasets have this size), you have to map the feature to a similar scale of dimension, which is computationally intractable and introduces many additional overhead. This limitation largely restricts the application of the proposed method.

- Some important references are missing from the paper. What the simplex is actually doing is to encourage hyperspherical uniformity. This have a large body of works on this, e.g. [1], [2], [3]. I think they are closely related, since the simplex classifier is essentially a special case of hyperspherical uniformity (I understand that the construction procedure could vary).

- The empirical gain seems to be somewhat marginal, especially on closed-set recognition.

[1] Learning towards Minimum Hyperspherical Energy, NeurIPS 2018

[2] Learning with Hyperspherical Uniformity, AISTATS 2021

[3] Regularizing Neural Networks via Minimizing Hyperspherical Energy, CVPR 2020

---

> ### Author Response · Authors · 2022-08-01
> **Response for Reviewer  a4BR**
>
> We would like to thank the reviewer for the nice comments given at the Strengths part.
>
> Weaknesses:
> Regarding the first weaknesses, the biggest limitation of the proposed method is its feature dimension restriction. We cannot apply it to datasets having very large number of classes without using dimension augmentation module. However, the number of classes is typically small especially in open set recognition tasks, therefore our proposed method is a perfect match for such problems and moderate sized datasets. Therefore, it can be still used in many classification problems.
> Regarding the second limitation, we will definitely add these references to the final version of the paper. Although these methods are related to our study, there are big differences: First of all, we just use the proposed loss function as the final layer for classification purposes. Therefore, our proposed method bears more similarity to UniformFace method [18] given in our paper.  This method also proposes a classification loss function for learning uniformly distributed representations on the hypersphere manifold through potential energy minimization as in [1,2,3]. However, the studies given at [1,2,3] consider the layer regularization problem and apply hyperspherical uniformity to the learned weights. Therefore, these methods are more complex (in some sense it is also more sophisticated since it applies the hyperspherical uniformity to all neural network layers). Consequently, there are many hyperparameters that must be fixed in the resulting method. In contrast, our proposed method is simple and there is no hyperparameter to tune. We also would like to point out the fact that these methods will have the same feature dimension restriction as in our proposed method if the methodology is applied to last classification layer. This can be seen in the arguments provided in [2]. More precisely, [2] demonstrates the close ties between the hyperspherical uniformity and orthogonality. In order to obtain an orthogonal (or orthonormal) set for d weight vectors, the dimension of the feature space must be higher than or equal to d. If there are one million classes for separation, one has to learn one million weight vectors for each class, which in return requires at least one million dimensional embedding space. Therefore, all methods given in [1,2,3] have the same dimension problem as in the proposed method.
> Lastly, we would like to point out that our proposed method significantly outperforms all these methods. For example, our error rate is 4.1% on Cifar10 and 20.5% on Cifar100 datasets. In contrast, the method proposed in [1] yields 6.21% error rate on Cifar10 and 25.61% error rate on Cifar100; the method proposed in [2] produces 20.97% error rate on Cifar100 dataset with a deeper network, and finally [3] results in 24.33% error rate on Cifar100 dataset.
> Regarding the empirical gain of closed set recognition, the gain is not significant for easy datasets such as Mnist and Cifar10, but the gain is very significant for Cifar-100 dataset. It is 3.4% better than the closest best performing method, (the method using the Center Loss).
>
> Questions:
> 1) For closed set recognition, we applied the simple loss function given in Eq. (4). Since the hypersphere is centered at the origin, it does not matter whether the angles or Euclidean distances are used for classification. Both metrics yield the same accuracy. ArcFace and all related method mentioned in the paper work well for face recognition problems, but they do not perform well on more general object classification problems where there are large intra-class variations. The reasons for this are given on page 2 (lines 85-92). More precisely, face class samples in specific classes can be approximated by using linear/affine spaces, and the similarities can be measured well by using the angles between sample vectors in such cases. However, the subspace approximation does not work for many general classification problems, such as Cifar-100 or ImageNet datasets, where there are large intra-class variations and the subspace approximations do not fit to the classes.
> 2) The feature dimension is 512 for the Cifar10 dataset since we use the ResNet-18 architecture. We did not use a 9-dimensional feature space. As class centers, we used the first 10 simplex vertices obtained by using the formulation given in Equations (1-2). Please note that, we do not need the reduce the dimension to the C-1 in order to use the proposed method. In Cifar10 dataset, we simply created 10 class centers whose dimension is 512.

---

> > ### Comment · Reviewer_a4BR · 2022-08-06
> > **Thanks for the response**
> >
> > Thanks for the response.
> >
> > 1. Whether open-set recognition tasks have large or small number of tasks still does not change the fact that this method is largely limited to small number of classes. I think this is in general a huge limitation for many applications (for example, face recognition).
> >
> > 2. [1] is in fact a superset of UniformFace. If I understand it correctly, its experiment on face recognition (sphereface+ as in the paper) is exactly to apply the uniform loss to the last layer. Please correct me if I misunderstood the paper.
> >
> > 3. The comparison to the other method may not be fair. I think there are more model parameters for the proposed method. It seems to be hard to evaluate where the experimental gain is from. It may be better to also show the number of model parameters in the experiment.

---

> > > ### Author Response · Authors · 2022-08-07
> > > **response for new comments**
> > >
> > > 1) We do not agree with the reviewer regarding the comment our proposed method is largely limited to small number of classes, and this is in general a huge limitation for many applications (for example, face recognition). The typical feature dimension is 2048 or 4096 in deep CNN architectures. Therefore, we can apply our proposed method to large datasets including 2049 or 4097 classes. For the most of the datasets considered as large-scale datasets, e.g., ImageNet, MS COCO, NUS-WIDE datasets, etc., the number of class categories is much smaller than these values. Therefore, we can apply our proposed to most of the large-scale classification problems without needing DAM module. The number of class categories is much larger than these values for mostly large-scale face recognition datasets. In such cases, we can use DAM module or our own architectures that do not use the last fully connected layers as we described before.
> > > 2) Regarding [1], we rechecked the paper again. As the reviewer stated, this method does not enforce the distances to be the same for all classes. Instead, it encourages uniform distributions on a hypersphere (in a way that the weights will not concentrate on certain areas and entire sphere shell will be used). This makes perfect sense especially for regularization of layer weights since the diverse weights will carry more information and reduce redundancy. When this idea is used as the final classification layer, the resulting method is similar to the UniformFace method as the reviewer pointed out. The authors also explicitly state that their method does not enforce orthogonality among the learned weights, otherwise it would be a completely different story and they would have the same restriction as in our proposed method. But, the same authors publish a more recent paper [2] recommended by the reviewer. In this paper, the authors show close ties between the hypersperical uniformity and orthogonality. More precisely, they show that orthogonal or orthonormal weight vectors are uniformly distributed on a hypersphere. As we indicated in our previous replies, any method that will enforce the orthogonality for hyperspherical uniformity will have the same dimension problem as in our proposed method. Lastly, please check the top of the page 5 in [2]. The authors state that the vertices of a regular (d + 1)-simplex (i.e., (d + 1)-dimensional convex hull of d+ 2 distinct vectors with equal pairwise distances) are universally optimal. This is exactly what we proposed in our paper. This clearly shows that our proposed method provides a global optimal solution for hyperspherical uniformity.
> > > 3) Regarding “The comparison to the other method may not be fair. I think there are more model parameters for the proposed method”, we are not sure what the reviewer meant with more model parameters. Which parameters are the reviewer referring to? For closed set recognition, we need to fix only u parameter and we already explained how we fixed this parameter. We also provided our ablation study results showing the accuracy changes based on different u values. For open set recognition, we need to fix m and \lambda values. Regarding m parameter, we did not have any trouble for fixing it since our centers are fixed to certain positions. We already know the distances between the class centers chosen as simplex vertices. All distances are equal, we simply checked the largest intra-class distances within classes and determined a margin based on this. Setting margin term to half of the radius worked well for all cases. For \lambda values, we fixed it based on cross-validation and come up with a general formulation for it.

---

> > > > ### Comment · Reviewer_a4BR · 2022-08-07
> > > > **Thanks for the response**
> > > >
> > > > I appreciate the authors for the their response.
> > > >
> > > > 1. The major concern that I have is that its feature dimension scales linearly with the number of classes, while the standard methods do not. For example, learnable classifiers can have 1024 feature dimension for million-class classification (meaning that the feature dimension is independent of the number of classes). This is the major limitation that I am talking about.
> > > >
> > > > 2. I have no doubt that the proposed method encourages hyperspherical unformity and can be viewed as a solution in (d+1 x d) cases. My point is just to clarify that the original response "[1] is different from UniformFace", since this contradicts what I understood.
> > > >
> > > > 3. There are multiple aspects for the additional parameters. First, the DAM module will inevitably consume more model parameters. Second, to obtain the same performance, it is likely that the standard training can do it without using that many parameters. I think this concern can be well addressed, if the authors can conduct closed-set experiments on more standard settings, say the same CIFAR-100 or ImageNet settings as [4] (also with reported number of model parameters). In that case, we are very familiar with what the performance of standard ResNet looks like.
> > > >
> > > > [4] Identity Mappings in Deep Residual Networks, ECCV 2016

---

> > > > > ### Author Response · Authors · 2022-08-08
> > > > > **response for new comments**
> > > > >
> > > > > 1) The reviewer is right, and most methods such as ArcFace or UniformFace, etc. do not have the feature dimension restriction as in our proposed method. However, many methods targeting uniform distributions [R1,R2,R3] on hyperspheres also have the same dimension restriction. But, we already proposed a method to handle this problem and it partially solved it. As we explained in our responses, we can provide a better solution if we change the network architectures.
> > > > >
> > > > > 2) Regarding our response “[1] is different from UniformFace”, sorry for the misunderstanding. We did not mean this and there is not such a comment in our response. We just wanted to emphasize that our proposed method is more similar to UniformFace method since it introduces a classification loss function as in our proposed method. On the other hand, [1] considers the layer regularization problem and apply hyperspherical uniformity to the learned weights in all layers. Therefore, this method is more complex (in some sense it is also more sophisticated since it applies the hyperspherical uniformity to all neural network layers). From this point of view, UniformFace is more like a special case of the [1] where hyperspectral uniformity is applied to only classification layer.
> > > > >
> > > > > 3) Using DAM module introduces more weights to learn as indicated by the reviewer. However, please note that we do not need DAM module for open and closed set recognition experiments conducted in our study. For closed set recognition experiments, architectures are identical for all tested methods (we used the same dimensional feature spaces for all tested methods), therefore they are directly comparable. We only used DAM module for face verification results given in Table 3.
> > > > >
> > > > > [R1] Do, Thanh-Toan, et al. "A theoretically sound upper bound on the triplet loss for improving the efficiency of deep distance metric learning." In CVPR. 2019.
> > > > >
> > > > > [R2] Nitin Bansal, Xiaohan Chen, and Zhangyang Wang. Can we gain more from orthogonality regularizations in training deep networks? In NeurIPS, 2018.
> > > > >
> > > > > [R3] Weiyang Liu, Yan-Ming Zhang, Xingguo Li, Zhiding Yu, Bo Dai, Tuo Zhao, and Le Song. Deep
> > > > > hyperspherical learning. In NIPS, 2017.

---

> > > > > > ### Comment · Reviewer_a4BR · 2022-08-08
> > > > > > **Thanks for the response**
> > > > > >
> > > > > > Thanks for the response. Despite some limitations, I think the paper proposes a simple yet useful idea. I hope my comments can help the authors improve their paper.

---

> > > > > > > ### Author Response · Authors · 2022-08-08
> > > > > > > **response**
> > > > > > >
> > > > > > > Thank you for constructive comments. Suggested related papers are highly appreciated and we will add those references and comparisons to these methods to the final version.

---

### Official Review · Reviewer_uWUv · 2022-07-28

**Rating:** 5
**Confidence:** 3
**Soundness:** 2 fair
**Presentation:** 3 good
**Contribution:** 2 fair

**Summary:**

Sorry for misunderstanding the paper. Revised review is here.

This paper proposed novel loss function for training deep neural network classifier, which can be generalizably applied to many domains such as evaluation for representation learning such as MoCo, BYOL or Masked Auto-Encoder. The loss function does not need any hyper-parameter tuning since the class centers are given as vertices of simplex and each data samples are trained to map to its corresponding class center. This way, traditional way of maximizing margin between samples from different classes and minimizing margin between samples from same class can be achieved in simple and effective way. While the limitation is the dimension of output feature of deep neural net should be larger or equal to C-1, authors mitigated the issue using DAM which can be further improved. In the experiment section, the authors gave intuition on how DSC clusters the data samples compared to other method, and how the learned representation can be semantically meaningful on unknown classes. The method achieved SOTA on most classification dataset which seems promising.

**Questions:**

1. Does the length of the center vector u affect the performance of the DSC (or need tuning)? How does this value affect the performance?
2. In equation (4), does the dimension of f_i, and s_{y_i} match? Since it seems that f_i is d-dimensional vector while s_{y_i} is C-dimensional vector. Please clarify this part.

**Limitations:**

None.

**Strengths And Weaknesses:**

strengths:
1. The paper is well-written and easy to follow.
2. The proposed method DSC generalizes to domains where the classification is required as a downstream task.
2. Proposed method achieves SOTA on various dataset

weaknesses:
1. More experiments on validating the proposed method seems required. For example, by getting learned representations from self-supervised pre-trained model (e.g., MoCo, BYOL, etc), the evaluation on downstream classification task with various loss functions will provide more strength on the proposed method.
2. As authors pointed out in the contribution, experimental evidence on how this method is robust to unbalanced dataset needs to be addressed.
3. The solution of limitation seems not practical, since simple multi-layer perceptron is used between d-dimensional and C-1 dimensional vector which requires a lot of parameters of d and C-1 is large.
4. It would be better to show distance matrix of other baselines. Although baselines' distance between class centers from known classes exhibit difference, semantic property would be preserved. Reviewer is not sure if the learned representation with DSC is actually semantically meaningful since this is not preserved for known classes. Giving more explanations on this would be nice.

---

> ### Author Response · Authors · 2022-07-30
> **Response for the first reviewer (Reviewer uWUv)**
>
> 1) Regarding weakness 1, we do not combine any existing methods (If the reviewer knows similar methods, we will be glad if he/she shares those methods with us). In contrast, we propose a novel deep neural network classifier loss function that enforces the samples of classes to cluster around the centers that are chosen from the vertices of a regular simplex. This is a completely new methodology. The existing methods we discussed in Section 2.1 (Motivation) simply show that the high-dimensional data concentrate on the vertices of a regular simplex. There is one method using this information for clustering, yet it is a traditional unsupervised machine learning method using hand-crafted features. Regarding motivation, we allocated a complete subsection (Section 2.1 Motivation) to explain our motivation. Please read this subsection again.
> 2) Claiming that classification task itself a small contribution is a complete nonsense. The classification is very important and active research area in machine learning and artificial intelligence fields. In fact, the reviewer admits this fact with his/her own words written in Limitations part. This is reviewer’s sentence: “The applicability of the proposed method is too limited. The recent trend is to learn good representations from the data and use those representations in various downstream tasks such as classification problems.” On one hand, the reviewer claims that classification is not very important , yet on the other hand he/she advises us a to apply our method for classification. We have done exactly the same thing. Our proposed method is a deep neural network classifier. We doubt that the reviewer have read a different paper.
> 3) Regarding the choice of the neural network architecture, we have proposed a novel classification loss function for deep neural networks and compared it to other state-of-the-art loss functions on the same architecture. Here, the main goal is to demonstrate the superiority of the proposed loss function over other loss functions. From this point of view, architecture type is irrelevant since we use the same architecture for all tested loss functions. Regarding comparison to other two methods, we can compare our proposed method to MoCo since it also proposes a similarity learning function that can be used for classification. But, comparing to BYOL is irrelevant since it introduces a novel architecture. In this study, we only propose a new classification loss function, not a different neural network architecture.
> Regarding Limitations, the reviewer contradicts with himself/herself. The reviewer advises us to show the use of the learned features for classification. This is what we have done in the paper. Regarding advantages of the proposed method, we clearly indicated them in the Contributions part of the paper. Here, we repeat them again since it seems the reviewer did not read the paper carefully. The advantages of the proposed method over existing methods can be summarized as follows:
> i) The proposed loss function does not have any hyper-parameter that must be fixed for classical
> classification problems, therefore it is extremely easy for the users. For open set recognition, the user has to set two parameters if the background class samples are used for learning.
> ii) The proposed method returns compact and interpretable acceptance regions for each class,
> thus it is very suitable for open set recognition problems.
> iii) The distances between the samples and their corresponding centers are minimized independently of each other, thus the proposed method also works well for unbalanced datasets.

---

> ### Author Response · Authors · 2022-08-02
> **Response for Reviewer uWUv**
>
> We would like to thank the reviewer for revision of his/her review with a more fair one.
>
> Weaknesses:
> 1) As we indicated in our first response, we have proposed a novel classification loss function for deep neural networks and compared it to other state-of-the-art loss functions on the same architecture. We did not propose a novel deep neural network architecture. Therefore, we can compare our proposed method to MoCo since it also proposes a similarity learning function that can be used for classification. But, comparing to BYOL is irrelevant since it introduces a completely novel architecture. We will add results of MoCo method in the final version.
> 2) Regarding robustness to unbalanced datasets, we conducted some simple tests on Cifar-10 dataset examples plotted in Fig. 3. In our proposed method, the distances between the samples and their corresponding centers are minimized independently of each other, thus the proposed method worked well and returned similar embeddings as in Fig. 3(a) even one of the class samples are significantly reduced. We will try to fit these results to Section 3.1 in the final version.
> 3) Regarding DAM module, we wanted to design a plug and play module that can be used with any desired deep CNN architecture without any changes. DAM module partially solved our problem and yielded satisfactory accuracies closer to the state-of-the-art. We can provide a better solution by designing our own architecture instead of using our proposed plug and play DAM module. To this end, we can avoid the fully connected layers that are used for dimension reduction in the last layers of deep CNNs. For example, in ResNet architectures, the dimension of the feature space is 25088 just before fully connected layers, and it is reduced to 512 after fully connected layers. We can avoid the last fully connected layers and use high-dimensional outputs of these earlier layers.  For example, using 25088 dimensional feature space is enough for training the large-scale MS1MV2 dataset we used in our tests without any need for dimension increase. At this point, we would like to point out the fact that all methods [1,2,3,4] (recommended by other reviewers) that target uniformly distributed class centers on the hypersphere have the same dimension problem. The authors simply did not realize it or they did not conduct experiments on large-scale as we did. Furthermore, [4] provided a wrong solution just using fully connected layers to increase the dimension.
> 4) Regarding distance matrix computation, we believe that there is no need to compute the distance matrices for other tested methods such as Softmax, ArcFace or CosFace since it is well known that they already return semantically meaningful embeddings. In our case, since the proposed method enforces the same distances between training classes, semantic relations are ignored among the training classes. We just wanted to demonstrate that our proposed method returns meaningful feature embeddings for open set recognition settings where there are unseen classes during testing stage. The distance matrix given in Fig. 4 demonstrates that the proposed method returns feature embeddings that respect semantic relations among the training classes and unknown test classes.
>
> Questions:
> 1) Experiments verify that the selection of u is not very important as long as it is not fixed to small values such as 1. Theoretically, the data samples lie on the surface of a growing hypersphere as the dimension increases. For smaller dimensions, we can choose smaller values of u as we did for illustrations experiments (we fixed u to 5 for 2-dimensional inputs). But, for larger dimensions we need higher values. Also, after some value, increasing u value does not change the results much. These are the accuracies we obtained for Cifar100 dataset for various u values:
>
> u=32, accuracy = 76.2%
>
> u=64, accuracy = 79.5%
>
> u= 100, accuracy = 79.4%
>
>  u= 150, accuracy = 79.9%
>
> u= 200, accuracy = 79.0%
>
> 2) The dimensions of the vectors match, both vectors come from d-dimensional feature space. There is no need to fix the dimension to C-1 to apply the proposed method. The proposed method can be used as long as d is larger than or equal to C-1.
>
> [1] Learning towards Minimum Hyperspherical Energy, NeurIPS 2018
>
> [2] Learning with Hyperspherical Uniformity, AISTATS 2021
>
> [3] Regularizing Neural Networks via Minimizing Hyperspherical Energy, CVPR 2020
>
> [4] Do, Thanh-Toan, et al. "A theoretically sound upper bound on the triplet loss for improving the efficiency of deep distance metric learning." In CVPR. 2019.

---

> > ### Comment · Reviewer_uWUv · 2022-08-08
> > **Thanks for the detailed response**
> >
> > The reviewer has read the rebuttal, and most of the concerns are resolved.
> >
> > The reviewer is not sure whether the authors understood the evaluation of the proposed method on representation learning settings. The point was to evaluate the learned representation from recent self-supervised learning method  such as BYOL or MoCo on some dataset (e.g., ImageNet) with proposed loss function (Since original MoCo or BYOL did not use the proposed loss function for downstream classification task), not directly comparing with MoCo or BYOL.

---

> > > ### Author Response · Authors · 2022-08-08
> > > **response for the new comments**
> > >
> > > Unfortunately, for both methods, we cannot use the learned representation from recent self-supervised learning method such as BYOL or MoCo on some dataset with proposed loss function. Actually, MoCo already uses a distance metric learning function called contrastive loss function (it is given in Eq. (1) in MoCo paper) for classification. As we explained at the Introduction part of our paper, contrastive loss minimizes the Euclidean distances between positive sample pairs and maximizes the distances between negative samples pairs. Therefore, it learns feature embeddings based on this contrastive loss function, and it is much better to compare this method directly to the proposed one in this case. Regarding BYOL, it uses two deep CNN architectures, referred to as online and target networks, that interact and learn from each other. BYOL simply starts from an augmented view of an image, and then it trains its online network to predict the target network’s representation of another augmented view of the same image. Therefore, it is hard to integrate our proposed loss function to such a specially designed network.

---

### Official Review · Reviewer_DKz4 · 2022-08-05

**Rating:** 5
**Confidence:** 3
**Soundness:** 2 fair
**Presentation:** 3 good
**Contribution:** 2 fair

**Summary:**

This paper proposed a method to combine the margin maximization in both Euclidean and angular spaces. More specifically, the method maps the labels to the vertices of a regular simplex as new labels.

Section 2:
1) explains the motivation of the method, i.e. the two distances are the same once the data lie on the boundary of a hyper-sphere.
2) more details about the method, including the vertices of a regular simplex, the case that uses the background class samples and the Dimension Augmentation Module(which is used for increasing the dimension, since the method requires the dimension of data is nearly the data size).

Section 3 and Appendix provide more details about the experiments:
1) feature representations learned by different loss function.
2) results of Open Set Recognition and Closed Set Recognition.


**Questions:**

1. Is there more explains about the hyperparameters such as, how there were chosen? Especially the u, which is the size of the hypersphere(it seems the u doesn't affect the classification?) . In contrast, is the m very sensitive? And what's the initialization?(what do the paper means by 'completely random').
2. Does DAM work better than the method in [1], or just the method in [1] is not suitable?

[1]Almost-equidistant sets.

**Limitations:**

No. The authors partly mentioned the limitations in the experiments (Section 3, especially 3.3.2) and the summary (Section 4). The limitation focus on the large-scale problem and the DAM.

**Strengths And Weaknesses:**

*The proposed method is interesting(also the DAM is interesting) and simple, and the simplicity brings several advantages (few hyper-parameters, proper acceptance regions, suit for unbalanced datasets), thus can be easily applied for classification problems. The experiments show that the method achieves good accuracy in several cases (especially the Open Set Recognition).
*The paper is well organized(especially the Section 2, which explains the main idea of the paper).

*Although the paper explains the benefits of the setting, it may need more explains or experiments or theorems to support the method. It's hard to convince me that the method works well now.
*The DAM seems need large size when the C is large.

---

> ### Author Response · Authors · 2022-08-05
> **Response for Weaknesses**
>
> First of all, we would like to correct a misunderstanding. We do not map any label to the vertices of a regular simplex (in fact, in the proposed method, we treat the labels as scalars, therefore such an approach does not make any sense in our setting. It may be possible if the labels are encoded as one hot vectors, but we do not treat them as vectors in the proposed method). Instead, we map the feature vectors of the samples in the classes to the vertices of the regular simplex where each class is approximated with a simplex vertex. The size of the feature samples is d, and it is required that it is larger than or equal to C-1. Also, all distances between the classes are same, not just the distances between two classes.
>
> Weaknesses:
> Regarding the weakness “Although the paper explains the benefits of the setting, it may need more explains or experiments or theorems to support the method. It's hard to convince me that the method works well now”, as we clearly indicated in our Motivation subsection in the paper, there are theoretical studies proving that the data samples lie at the vertices of a regular simplex in high-dimensional spaces. Therefore, a classifier mapping the class specific samples to the vertices of a simplex makes perfect sense. In addition, since the class samples are compactly clustered around their corresponding class centers, this makes the proposed method ideal for open set recognition tasks where one needs to reject the unknown class samples during testing phase. Experimental results also verify theoretical findings since the proposed method typically achieves the state-of-the-art accuracies as seen in Table 1. More precisely, although we proposed a general classification method for closed set recognition settings, the proposed method outperforms all existing sophisticated open set recognition methods. Our accuracies are now new state-of-the-art accuracies on most of the tested datasets. Moreover, our proposed method also beats related state-of-the-art loss functions on closed set recognition experiments. The performance difference is significant especially on Cifar-100 dataset as seen in Table 2. Our proposed method could not beat the state-of-the-art only for large-scale face recognition problems since we had to use DAM module in these experiments. However, our accuracies are still encouraging in the sense that they are generally closer to the best reported accuracies.
> Regarding DAM module, as we explained for other reviewers, we wanted to design a plug and play module that can be used with any desired deep CNN architecture without any changes. DAM module partially solved our problem and yielded satisfactory accuracies closer to the state-of-the-art. We can provide a better solution by designing our own architecture instead of using our proposed plug and play DAM module. To this end, we can avoid the fully connected layers that are used for dimension reduction in the last layers of deep CNNs. For example, in ResNet architectures, the dimension of the feature space is 25088 just before fully connected layers, and it is reduced to 512 after fully connected layers. We can avoid the last fully connected layers and use high-dimensional outputs of these earlier layers.  For example, using 25088 dimensional feature space is enough for training the large-scale MS1MV2 dataset we used in our tests without any need for dimension increase. At this point, we would like to point out the fact that all methods that target uniformly distributed class centers on the hypersphere have the same dimension problem. The authors simply did not realize it or they did not conduct experiments on large-scale as we did.

---

> ### Author Response · Authors · 2022-08-05
> **Response for Questions**
>
> Here we answer the questions raised by the reviewer. For Weaknesses part, see our first response.
>
> Question:
> 1) Regarding parameters, selection of u is not very important as long as it is not fixed to small values such as 1. Theoretically, the data samples lie on the surface of a growing hypersphere as the dimension increases. For smaller dimensions, we can choose smaller values of u as we did for illustrations experiments (we fixed u to 5 for 2-dimensional inputs). But, for larger dimensions we need higher values. Also, after some value, increasing u value does not change the results much. These are the accuracies we obtained for Cifar100 dataset for various u values:
>
> u=32, accuracy = 76.2%
>
> u=64, accuracy = 79.5%
>
> u= 100, accuracy = 79.4%
>
>  u= 150, accuracy = 79.9%
>
> u= 200, accuracy = 79.0%
>
> Regarding m parameter, it is only used for open set recognition problems. Moreover, we did not have any trouble for fixing it since our centers are fixed to certain positions. We already know the distances between the class centers chosen as simplex vertices. All distances are equal, we simply checked the largest intra-class distances within classes and determined a margin based on this. Setting margin term to half of the radius worked well for all cases. For all experiments, we did not fine-tune our classification network from a pre-trained network and started the network weights from scratch by initializing with random weights which is the common practice used for initializing network weights.
>
> 2) Regarding [1], it does not directly solve the problem. It simply allows to use 2d+4 class centers instead of d+1 centers in d-dimensional spaces. If 2d+4 is smaller than C, we cannot use it for classification. In contrast, DAM module allows us to increase the dimension of the feature space to any desired number without any restriction.

---

### Meta-Review · Area_Chair_tr8q · 2022-08-24

**Recommendation:** Reject
**Confidence:** Certain

**Metareview:**

This paper proposed to use least-squares loss functions in training deep neural networks. The main idea is to encode class means, whose mutual distances are equivalent. The method is simple but efficient. However, the similar idea has been widely used in multi-class classification (SVM and Fisher discriminant analysis) and spectral clustering. More specifically,  one reviewer commented that this work encodes class labels as high-dimensional vectors  similar one-hot, and then uses a least-squares loss. Although the authors did not admit this comment, but essecially this comment is indeed right. This idea has been used such as in the following references
1) Multicategory Support Vector Machines: Theory and Application to the Classification of Microarray Data and Satellite Radiance Data
Yoonkyung Lee, Yi Lin & Grace Wahba
2) Prevalence of neural collapse during the terminalphase of deep learning training  Vardan Papyana, X. Y. Hanb, and David L. Donoho


**Award:**

No

---

### Decision · Program_Chairs · 2022-09-14

Reject